# Atherosclerotic cardiovascular disease thresholds for statin initiation among people living with HIV in Thailand: A cost-effectiveness analysis

David C. Boettiger[1,2,3‡]*, Pairoj Chattranukulchai[4‡], Anchalee Avihingsanon[5], Romanee Chaiwarith[6], Suwimon Khusuwan[7], Matthew G. Law[1], Jeremy Ross[8], Sasisopin Kiertiburanakul[9]

**1** Kirby Institute, UNSW Sydney, Sydney, Australia, **2** Institute for Health and Aging, University of California, San Francisco, California, United States of America, **3** Faculty of Medicine, Biostatistics Excellence Centre, Chulalongkorn University, King Chulalongkorn Memorial Hospital, Bangkok, Thailand, **4** Faculty of Medicine, Division of Cardiovascular Medicine, Chulalongkorn University, Bangkok, Thailand, **5** HIV-NAT Collaboration/Thai Red Cross AIDS Research Centre, Bangkok, Thailand, **6** Research Institute for Health Sciences, Chiang Mai University, Chiang Mai, Thailand, **7** Chiangrai Prachanukroh Hospital, Chiang Rai, Thailand, **8** TREAT Asia/amfAR–Foundation for AIDS Research, Bangkok, Thailand, **9** Faculty of Medicine Ramathibodi Hospital, Mahidol University, Bangkok, Thailand

‡ These authors share first authorship on this work.
* dboettiger@kirby.unsw.edu.au

**Data Availability Statement:** These data were collected as part of a regional cohort collaboration. The cohort collaboration has data-sharing policies

## Abstract

### Background

People living with HIV (PLHIV) have an elevated risk of atherosclerotic cardiovascular disease (ASCVD) compared to their uninfected peers. Expanding statin use may help alleviate this burden. We evaluated the cost-effectiveness of reducing the recommend statin initiation threshold for primary ASCVD prevention among PLHIV in Thailand.

### Methods

Our decision analytic microsimulation model randomly selected (with replacement) individuals from the TREAT Asia HIV Observational Database (data collected between 1/January/ 2013 and 1/September/2019). Direct medical costs and quality-adjusted life-years were assigned in annual cycles over a lifetime horizon and discounted at 3% per year. We assumed the Thai healthcare sector perspective. The study population included PLHIV aged 35–75 years, without ASCVD, and receiving antiretroviral therapy. Statin initiation thresholds evaluated were 10-year ASCVD risk ≥10% (control), ≥7.5% and ≥5%.

### Results

A statin initiation threshold of ASCVD risk ≥7.5% resulted in accumulation of 0.015 additional quality-adjusted life-years compared with an ASCVD risk threshold ≥10%, at an extra cost of 3,539 Baht ($US113), giving an incremental cost-effectiveness ratio of 239,000 Baht ($US7,670)/quality-adjusted life-year gained. The incremental cost-effectiveness ratio

that were approved by the corresponding IRB and specify that both internal and external investigators are subject to a formal process to request access to the data through submission of a concept sheet that adheres to these policies. This study was conducted under these policies, and data will only be available upon request for researchers who meet the criteria for access to confidential data. Interested individuals should contact Boondarika Petersen (tor.nakornsri@treatasia.org).

**Funding:** No funding was received for the modelling analysis presented. The TREAT Asia HIV Observational Database is an initiative of TREAT Asia, a programme of amfAR, The Foundation for AIDS Research, with support from the U.S. National Institutes of Health's National Institute of Allergy and Infectious Diseases, the Eunice Kennedy Shriver National Institute of Child Health and Human Development, the National Cancer Institute, the National Institute of Mental Health, the National Institute on Drug Abuse, the National Heart, Lung, and Blood Institute, the National Institute on Alcohol Abuse and Alcoholism, the National Institute of Diabetes and Digestive and Kidney Diseases, and the Fogarty International Center, as part of the International Epidemiology Databases to Evaluate AIDS (IeDEA; U01AI069907). The Kirby Institute (data center for the TREAT Asia HIV Observational Database) is funded by the Australian Government Department of Health and Ageing, and is affiliated with the Faculty of Medicine, UNSW Sydney. The content of this publication is solely the responsibility of the authors and does not necessarily represent the official views of any of the governments or institutions mentioned above.

**Competing interests:** DCB has received research funding from Gilead Sciences and is supported by a National Health and Medical Research Council Early Career Fellowship (APP1140503); MGL has received unrestricted grants from Boehringer Ingelhiem, Gilead Sciences, Merck Sharp & Dohme, Bristol-Myers Squibb, Janssen-Cilag, and ViiV HealthCare and consultancy fees from Gilead Sciences and data and safety monitoring board sitting fees from Sirtex Pty Ltd; All other authors report no potential competing interests. These declarations do not alter our adherence to PLOS ONE policies on sharing data and materials.

**Abbreviations:** ART, Antiretroviral therapy; ASCVD, Atherosclerotic cardiovascular disease; CABG, Coronary artery bypass graft; D:A:D, Data-collection on Adverse Effects of Anti-HIV Drugs; ICER, Incremental cost-effectiveness ratio; LDL-C, Low density lipoprotein cholesterol; PCI, Percutaneous coronary intervention; PLHIV, People

comparing ASCVD risk $\geq$5% to $\geq$7.5% was 349,000 Baht ($US11,200)/quality-adjusted life-year gained. At a willingness-to-pay threshold of 160,000 Baht ($US5,135)/quality-adjusted life-year gained, a 30.8% reduction in the average cost of low/moderate statin therapy led to the ASCVD risk threshold $\geq$7.5% becoming cost-effective compared with current practice.

## Conclusions

Reducing the recommended 10-year ASCVD risk threshold for statin initiation among PLHIV in Thailand would not currently be cost-effective. However, a lower threshold could become cost-effective with greater preference for cheaper statins.

## Introduction

People living with HIV (PLHIV) have an elevated risk of atherosclerotic cardiovascular disease (ASCVD) compared to their uninfected peers [1]. Causes of this excess ASCVD risk are a poorly understood combination of immune deficiency, antiretroviral therapy (ART) use, co-infections such as hepatitis B and C, and lifestyle factors such as cigarette smoking and alcohol use [2].

Statins reduce ASCVD risk by lowering low-density lipoprotein cholesterol (LDL-C) levels [3]. It has also been hypothesized that the anti-inflammatory properties of statins, including reductions in soluble CD14, oxidized LDL-C, and lipoprotein-associated phospholipase 2 [4], may further reduce ASCVD risk in PLHIV [5]. Current Thai guidelines recommend statin therapy for primary ASCVD prevention among PLHIV with a 10-year ASCVD risk $\geq$10% [6], consistent with general population guidelines [7]. Reducing the risk threshold for statin initiation in PLHIV may help to alleviate their excess burden of ASCVD.

We recently reported that it would not be cost-effective to expand pravastatin or pitavastatin use to PLHIV in Thailand not currently on lipid-lowering therapy [8]. Pravastatin and pitavastatin are preferred statins in the context of HIV due to their lack of interaction with ART. However, many prescribers opt for cautious use of other statins due to their low cost. We therefore aimed to determine the cost-effectiveness of lowering the recommended ASCVD risk threshold for statin initiation among PLHIV in Thailand, assuming current statin prescribing patterns.

## Methods

### Ethics statement

Ethics approval was granted for the TAHOD study design, methods and consent procedures by the University of New South Wales Human Research Ethics Committee (HC17825). Site specific study governance was granted by site-relevant institutional review boards: Ministry of Health National Ethics Committee for Health Research (Cambodia), Ethical Committee of Beijing Ditan Hospital Affiliated to Capital Medical University (China), Research Ethics Committee Kowloon Central / Kowloon East, Hospital Authority IRB (China), Institutional Review Board Of YRG CARE (India), Institutional Ethics Committee Rao Nursing Home (India), Kerti Praja Foundation IRB (Indonesia), Committee of Medical Research Ethics, Faculty of Medicine University of Indonesia (Indonesia), National Center for Global Health and Medicine Human Research Ethics Committee (Japan), Medical Research & Ethics Committee,

living with HIV; QALY, Quality-adjusted life-year;
Rama-EGAT, Ramathibodi-Electricity Generating
Authority of Thailand; T1MI, Myocardial infarction;
TAHOD, TREAT Asia HIV Observational Database.

Ministry of Health (for Sungai Buloh Hospital and Hospital Raja Perempuan Zainab II, Malaysia), Medical Ethics Committee, University Malaya Medical Centre (Malaysia), Research Institute for Tropical Medicine, Department of Health (Philippines), National Healthcare Group IRB, Domain Specific Review Board (Singapore), Severance Hospital Yonsei University College of Medicine Institutional Review Board (South Korea), Institutional Review Board of Taipei Veterans General Hospital (Taiwan), The Internal Ethical Committee for Research in Human Subject, Chiangrai Prachanukroh Hospital (Thailand), Institutional Review Board Faculty of Medicine, Chulalongkorn University (Thailand), Committee on Human Rights Related to Research Involving Human Subjects Faculty of Medicine Ramathibodi Hospital, Mahidol University (Thailand), Research Ethics of the Faculty of Medicine, Chiang Mai University (Thailand), Siriraj Institutional Review Board, Mahidol University (Thailand), Ministry of Health, Hanoi School of Public Health IRB (Vietnam), and National Hospital of Tropical Diseases IRB (Vietnam).

Written informed consent was not sought in TAHOD unless required by a site's local institutional review board. The need for written consent was waived by the following ethics committees: Ministry of Health National Ethics Committee for Health Research (Cambodia), Ethical Committee of Beijing Ditan Hospital Affiliated to Capital Medical University (China), Research Ethics Committee Kowloon Central / Kowloon East, Hospital Authority IRB (China), Institutional Review Board Of YRG CARE (India), Institutional Ethics Committee Rao Nursing Home (India), Kerti Praja Foundation IRB (Indonesia), Committee of Medical Research Ethics, Faculty of Medicine University of Indonesia (Indonesia), Medical Research & Ethics Committee, Ministry of Health (for Sungai Buloh Hospital, Malaysia), Medical Ethics Committee, University Malaya Medical Centre (Malaysia), The Internal Ethical Committee for Research in Human Subject, Chiangrai Prachanukroh Hospital (Thailand), Ministry of Health, Hanoi School of Public Health IRB (Vietnam), and National Hospital of Tropical Diseases IRB (Vietnam).

### Study population

We used individual, de-identified patient data from Thai sites contributing to the TREAT Asia HIV Observational Database (TAHOD). TAHOD involves 21 HIV clinics in the Asia-Pacific region and is part of the International Epidemiology Databases to Evaluate AIDS collaboration [9]. Ethics approval was granted by the University of New South Wales Human Research Ethics Committee (HC17825). Clinics follow local guidelines and regulations regarding patient consent and ethics review (see Ethics statement in Acknowledgements). The study population was selected from patients enrolled at Ramathibodi Hospital, Bangkok; HIV-NAT Research Collaboration/Thai Red Cross AIDS Research Centre, Bangkok; Research Institute for Health Sciences, Chiang Mai; or Chiangrai Prachanukroh Hospital, Chiang Rai. Included patients were required to have documentation of ≥1 visit between 1 January 2013 and 1 September 2019 and, at their last clinic visit, be aged 35–75 years, have no history of ASCVD, have been using ART for at least 6 months, and have a CD4 cell count >100 cells/mm$^3$. S1 Table in S1 File further characterizes the PLHIV in our study population.

### Model structure, parameterization and validation

We developed a microsimulation model that randomly selected (with replacement) 100,000 patients from our study population (N = 1,379). The model simulated their probability of ASCVD over time. Direct medical costs and outcomes were assigned in annual cycles over a lifetime horizon and discounted at 3% per year [10]. We assumed the Thai healthcare sector perspective. Primary ASCVD risk was calculated using the reduced Data-collection on Adverse

Effects of Anti-HIV Drugs (D:A:D) CVD risk equation [11]. Background mortality rates were based on those of the Asian population on ART [12]. Recurrent event rates were modelled as per our earlier analysis evaluating pravastatin and pitavastatin expansion [8]. Further detail on the model is provided in the Supplementary Material, including S1 Fig in S1 File which presents a schematic of the model structure. Model parameters are shown in Table 1. We calibrated our model using a goodness-of-fit approach based on the observed rates of all-cause and cardiovascular death among TAHOD participants between 2009 and 2019. Fig 1 shows that our calibrated model estimates provided an accurate reflection of the observed data.

## Treatment strategies

Current Thai guidelines recommend PLHIV with an LDL-C level ≥190mg/dL, those aged ≥40 years with diabetes, and those aged ≥50 years with chronic kidney disease and an LDL-C level ≥100mg/dL should be started on a statin regardless of ASCVD risk score [6,7]. For those who do not meet these criteria, statin therapy is recommended if their 10-year ASCVD risk is ≥10%. Individuals with an ASCVD score <10% may be considered for statin therapy if there is evidence of subclinical atherosclerosis but such information is rarely available in routine practice in Thailand. Consistent with Thai guidelines, we assumed all individuals with an LDL-C level ≥190mg/dL would be prescribed high intensity statin therapy, and those with a statin indication associated with diabetes or chronic kidney disease would be prescribed low/moderate intensity statin therapy. We evaluated three different treatment strategies for individuals who currently require an ASCVD risk score to determine statin eligibility: 1) treating those with a 10-year ASCVD risk ≥10% with a low/moderate intensity statin (control group); 2) treating those with a 10-year ASCVD risk ≥7.5% with a low/moderate intensity statin; and 3) treating those with a 10-year ASCVD risk ≥5% with a low/moderate intensity statin.

We modelled the effectiveness of statins through the simulated change in LDL-C. We assumed fully adherent individuals using high intensity and low/moderate intensity statins would achieve LDL-C reductions of 55% and 40%, respectively [3,13]. Statin adherence was assumed to be 86.7% in the first year of statin use, 72.6% in the second, 61.1% in the third, 56.6% in the fourth, and 58.1% in all years thereafter [14]. We assumed PLHIV would only accrue the cost of statin use, exhibit side effects of statins, and benefit from statin LDL-C cholesterol reduction while they were using a statin and hence these parameters were adjusted in line with the decline in adherence over time.

We assumed statin therapy only reduced ASCVD risk by improving LDL-C levels. In scenario analyses, we assumed additional ASCVD preventative efficacy to account for the possibility that the anti-inflammatory properties of statins provide additional benefit in PLHIV [5]. We assumed statins do not prevent non-ASCVD events as current evidence suggests little or no benefit for such outcomes [33]. We modelled adverse events related to statin use (hemorrhagic stroke, diabetes, and myopathy) based on rates observed in the general population (see Table 1 and Supplementary Material in S1 File). We did not account for differences in statin type or dose as current evidence suggests these factors have little impact on the type or frequency of adverse events observed [34].

## Cost and quality-of-life estimates

Health-related costs and quality-of-life adjustments were assigned to clinical events and health states in annual cycles. We included medical costs regardless of who paid for them. Cost estimates from earlier years were inflated to 2018 Thai Baht equivalents [35]. The cost of HIV management and rates of second-line ART use were based on published literature [15,16]. Statin costs were estimated based on unit costs [25] and a survey of statin use among PLHIV

**Table 1. Key model parameters.**

| Parameter | Base case (range for sensitivity) | Source |
|---|---|---|
| *Probabilities* | | |
| Probability of ASCVD event | Varies by individual based on D:A:D equation[a] | [11] |
| Probability of non-CVD death | Varies by age, sex and CD4[b] | [12] |
| *Statin efficacy* | | |
| Reduction in LDL-C associated with high intensity statin, % | 55.0 (50.0–60.0) | [3,13] |
| Reduction in LDL-C associated with low-moderate intensity statin, % | 40.0 (25.0–49.0) | [3,13] |
| Statin adherence | Varies by duration of statin use (+/- 15% of base-case) | [14] |
| *Costs, 2018 Thai Baht* | | |
| HIV management | 59,856 (29,929–89,784) | [15,16] |
| Non-fatal T1MI medical management[b] | 35,441 (17,721–53,162) | [17] |
| PCI[b] | 215,765 (107,882–323,647) | [17] |
| CABG[b] | 316,475 (158,238–474,714) | [17] |
| Non-fatal T1MI management—First year post-T1MI[b] | 62,245 (34,974–143,252) | [18] |
| Non-fatal T1MI management—After first year post-T1MI[b] | 17,780 (8,890–26,670) | [18] |
| Fatal T1MI[b] | 221,915 (81,878–356,072) | [18] |
| Non-fatal ischemic stroke hospitalization[b] | 26,668 (23,497–29,820) | [19,20] |
| Non-fatal ischemic stroke management—First year post-stroke[b] | 42,435 (39,284–45,587) | [19,20] |
| Non-fatal ischemic stroke management—After first year post-stroke[b] | 10,932 (8,746–13,119) | [19] |
| Fatal ischemic stroke[b] | 54,671 (43,737–65,606) | [19] |
| Non-fatal hemorrhagic stroke hospitalization[b] | 26,668 (23,497–29,820)[c] | Assumption |
| Non-fatal hemorrhagic stroke management—First year post-stroke[b] | 42,435 (39,284–45,587)[c] | Assumption |
| Non-fatal hemorrhagic stroke management—After first year post-stroke[b] | 10,932 (8,746–13,119)[c] | Assumption |
| Fatal hemorrhagic stroke[b] | 54,671 (43,737–65,606)[c] | Assumption |
| Other cardiovascular death[b] | 221,915 (81,878–356,072)[d] | Assumption |
| Statin-associated diabetes, average cost/individual taking statin/year[b] | 2.30 (1.70–3.70) | [21,22] |
| Statin-associated myopathy, average cost/individual taking statin/year[b] | 0.05 (0.02–0.08) | [23,24] |
| High intensity statin, 12-month supply | 9,486 (4,743–14,229) | [25] and site survey[e] |
| Low/moderate intensity statin, 12-month supply | 2,301 (1,151–3,452) | [25] and site survey[e] |
| *Utility weights* | | |
| No history of CVD | 1.0000 | Assumption |
| History of T1MI[b] | 0.7780 (0.6613–0.9758) | [26–29] |
| History of ischemic stroke[b] | 0.7680 (0.6528–0.9108) | [26–29] |
| History of hemorrhagic stroke[b] | 0.6100 (0.4000–0.8000) | [30] |
| *Quality-of-life decrements* | | |
| PCI[b] | 0.0061 (0.0040–0.0087) | [31] |
| CABG[b] | 0.0128 (0.0084–0.0184) | [31] |
| Acute T1MI[b] | 0.0076 (0.0051–0.0106) | [31] |

(*Continued*)

**Table 1.** (Continued)

| Parameter | Base case (range for sensitivity) | Source |
|---|---|---|
| Acute ischemic stroke[b] | 0.0242 (0.0158–0.0335) | [31] |
| Acute hemorrhagic stroke[b] | 0.0242 (0.0158–0.0335) | [31] |
| Diabetes, average quality-of-life decrement/individual taking statin/year[b] | 0.00005 (0.00003–0.00007) | [22,31] |
| Myopathy, average quality-of-life decrement/individual taking statin/year[b] | 0.0000010 (0.0000007–0.0000012) | [23,31] |
| Daily statin administration/pill burden[b] | 0.00000 (0.00000–0.00384) | [32] |
| *Discounting and time horizon* | | |
| Annual discount rate, % (applied to costs and benefits) | 3.0 (0.0–5.0) | [10] |
| Time horizon | Lifetime | [10] |

Baht can be converted to $US by dividing by 31.16

[a] D:A:D equation uses age, sex, diabetes status, family history of CVD, current and past smoking status, total cholesterol, high density lipoprotein cholesterol, systolic blood pressure, and CD4 cell count to calculate CVD risk

[b] Based on general population or high-income setting

[c] As for ischemic stroke hospitalization/management

[d] As for fatal T1MI. ASCVD, atherosclerotic cardiovascular disease

[e] Statin costs were estimated based on published unit costs and a survey of statin use among people living with HIV on antiretroviral therapy at sites contributing to our study population. D:A:D, Data-collection on Adverse Effects of Anti-HIV Drugs study; LDL-C, low density lipoprotein cholesterol; T1MI, type 1 myocardial infarction; PCI, percutaneous coronary intervention; CABG, coronary artery bypass graft.

on ART at the Thai TAHOD sites contributing to our study population. S4 Table in S1 File provides a breakdown of how we arrived at the average annual costs for low/moderate and high intensity statin use (2,301 Baht [$US74] and 9,486 Baht [$US304], respectively). Other costs were based on published estimates for the general population (see Table 1). Quality-of-life adjustments were largely based on data from the 2017 Global Burden of Disease study [31]. Since patients using ART already take at least one daily pill, our base-case model assumed the inconvenience of taking a daily statin (pill burden) was not associated with a quality-of-life decrement. Earlier studies among the HIV and general populations have also assumed regular statin use is not associated with a pill burden [8,32,36].

## Outcomes

The primary outcome was the incremental cost-effectiveness ratio (ICER). The threshold for an intervention being considered cost-effective (willingness-to-pay threshold) was as an ICER below 160,000 Baht ($US5,315). This is consistent with recommendations from the Thai Ministry of Public Health [37].

## Scenario analyses

In addition to our base-case analyses, we investigated the following scenarios:

1. Assuming statins reduce ASCVD event probability an additional 15% to account for the possibility that their anti-inflammatory properties provide additional benefit among PLHIV;

2. As above but assuming statins reduce ASCVD event probability an additional 30%;

3. Using the Rama-EGAT equation to calculate T1MI and ischemic stroke risk. The Rama-EGAT equation [38] has been validated in the general Thai population [39] and is a reasonable alternative to the D:A:D equation.

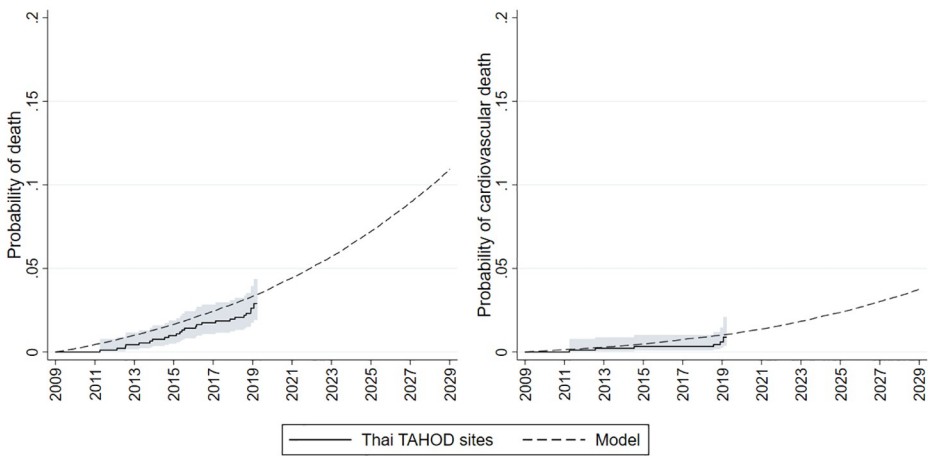

**Fig 1. Observed versus modelled probability of all-cause and cardiovascular death over time.** Observed data is from Thai sites in the TREAT Asia HIV Observational Database (TAHOD). Shaded area is 95% confidence interval for observed data.

## Sensitivity analyses

We used sensitivity analyses to evaluate whether our results were impacted by uncertainty in key input parameters. In deterministic sensitivity analyses, we varied one or two input parameters at a time. In probabilistic sensitivity analyses we varied multiple input parameters simultaneously across prespecified distributions over 1,000 iterations. We used beta distributions for utilities and event probabilities, and log-normal distributions for hazard ratios, safety and efficacy measures, and costs.

## Software

Data management and statistical analysis was conducted using SAS 9.4 (SAS Institute Inc, Cary, North Carolina). Modelling was undertaken in TreeAge Pro 2020 Version R1.0 (Tree-Age Software, Williamstown, Massachusetts).

## Results

### Base-case analysis

Modelled incidence rates for T1MI, ischemic stroke and fatal CVD among the ≥10% ASCVD risk threshold group were 6.91, 3.06 and 3.58 per 1,000 person-years, respectively. Individuals in this group were projected to accumulate a discounted average of 22.268 QALYs, 22.587 life-years and 1,446,067 Baht ($US46,407) in direct medical costs (Table 2).

Compared with the ≥10% ASCVD risk threshold group, reductions in the incidence of T1MI, ischemic stroke and fatal CVD were projected to be 0.9%, 1.3% and 0.7%, respectively, in the ≥7.5% ASCVD risk threshold group. These reductions contributed to the accumulation of 0.015 additional QALYs at an incremental cost of 3,539 Baht ($US113), giving an ICER of 239,000 Baht ($US7,670)/QALY gained. Compared with the ≥7.5% ASCVD risk threshold group, reductions in the incidence of T1MI, ischemic stroke and fatal CVD were projected to be 0.4%, 0.8% and 0.8%, respectively, in the ≥5% ASCVD risk threshold group. These reductions contributed to the accumulation of 0.015 additional QALYs at an incremental cost of 5,381 Baht ($US172), giving an ICER of 349,000 Baht ($US11,200)/QALY gained (Table 2).

**Table 2. Incremental cost-effectiveness of different statin initiation thresholds for primary prevention of ASCVD among PLHIV.**

| Intervention | Total cost, Baht | Statin cost, Baht | T1MI[a] | Ischemic stroke[a] | Fatal CVD[a] | Life-years | QALYs | Incremental cost, Baht | Incremental life-years gained | Incremental QALYs gained | Baht/life-year gained[b] | ICER, Baht/QALY gained[b] |
|---|---|---|---|---|---|---|---|---|---|---|---|---|
| *Base-case* | | | | | | | | | | | | |
| ≥10% ASCVD risk | 1,446,067 | 46,095 | 6.91 | 3.06 | 3.58 | 22.587 | 22.268 | - | - | - | - | - |
| ≥7.5% ASCVD risk | 1,449,606 | 49,910 | 6.85 | 3.02 | 3.55 | 22.595 | 22.283 | 3,539 | 0.007 | 0.015 | 476,000 | 239,000 |
| ≥5% ASCVD risk | 1,454,987 | 55,155 | 6.82 | 3.00 | 3.52 | 22.608 | 22.298 | 5,381 | 0.013 | 0.015 | 417,000 | 349,000 |
| *Scenario 1) Statins reduce ASCVD risk an additional 15% due to anti-inflammatory effects* | | | | | | | | | | | | |
| ≥10% ASCVD risk | 1,445,713 | 46,377 | 6.58 | 2.83 | 3.42 | 22.619 | 22.314 | - | - | - | - | - |
| ≥7.5% ASCVD risk | 1,449,144 | 50,191 | 6.52 | 2.79 | 3.40 | 22.627 | 22.329 | 3,431 | 0.008 | 0.015 | 450,000 | 231,000 |
| ≥5% ASCVD risk | 1,454,566 | 55,448 | 6.49 | 2.75 | 3.36 | 22.641 | 22.347 | 5,422 | 0.014 | 0.018 | 381,000 | 299,000 |
| *Scenario 2) Statins reduce ASCVD risk an additional 30% due to anti-inflammatory effects* | | | | | | | | | | | | |
| ≥10% ASCVD risk | 1,445,200 | 46,693 | 6.02 | 2.68 | 3.21 | 22.649 | 22.361 | - | - | - | - | - |
| ≥7.5% ASCVD risk | 1,448,858 | 50,521 | 5.96 | 2.62 | 3.17 | 22.662 | 22.381 | 3,658 | 0.013 | 0.021 | 286,000 | 177,000 |
| ≥5% ASCVD risk | 1,454,103 | 55,794 | 5.91 | 2.59 | 3.14 | 22.677 | 22.402 | 5,245 | 0.016 | 0.020 | 336,000 | 256,000 |
| *Scenario 3) Using Rama-EGAT equation* | | | | | | | | | | | | |
| ≥10% ASCVD risk | 1,459,913 | 52,768 | 8.36 | 3.69 | 4.03 | 22.584 | 22.216 | - | - | - | - | - |
| ≥7.5% ASCVD risk | 1,462,771 | 55,434 | 8.33 | 3.69 | 4.01 | 22.593 | 22.226 | 2,858 | 0.009 | 0.011 | 307,000 | 272,000 |
| ≥5% ASCVD risk | 1,466,379 | 58,934 | 8.34 | 3.68 | 4.00 | 22.599 | 22.235 | 3,608 | 0.006 | 0.009 | 566,000 | 424,000 |

Incremental cost-effectiveness for each strategy was measured relative to the next best strategy in terms of QALYs gained. Costs, QALYs, and life-years were discounted at 3%/year. Baht can be converted to $US by dividing by 31.16; T1MI, type 1 myocardial infarction; ASCVD, atherosclerotic cardiovascular disease; QALY, quality-adjusted life-year; ICER, incremental cost-effectiveness ratio; Rama-EGAT, Ramathibodi-Electricity Generating Authority of Thailand.

[a] per 1,000 person-years

[b] Rounded to nearest thousand.

## Scenario analyses

Our scenario analyses results are shown in Table 2. Assuming statins reduce ASCVD risk an additional 15% due to their anti-inflammatory effects (Scenario 1) reduced the ICERs for both the ≥7.5% ASCVD risk threshold versus the ≥10% ASCVD risk threshold (231,000 Baht [$US7,413]/QALY gained), and for the ≥5% ASCVD risk threshold versus the ≥7.5% ASCVD risk threshold (299,000 Baht [$US9,595]/QALY gained). Assuming statins reduce ASCVD risk an additional 30% due to their anti-inflammatory effects (Scenario 2) further reduced the respective ICERs: 177,000 Baht ($US5,680)/QALY gained and 256,000 Baht ($US8,215)/QALY gained. When replacing the D:A:D equation with the Rama-EGAT equation to estimate ASCVD risk (Scenario 3), our model predicted higher incidence rates of T1MI, ischemic stroke and fatal CVD. The ICERs were 272,000 Baht ($US8,729)/QALY gained for the ≥7.5% ASCVD risk threshold versus the ≥10% ASCVD risk threshold, and 424,000 Baht ($US13,607)/QALY gained for the ≥5% ASCVD risk threshold versus the ≥7.5% ASCVD risk threshold.

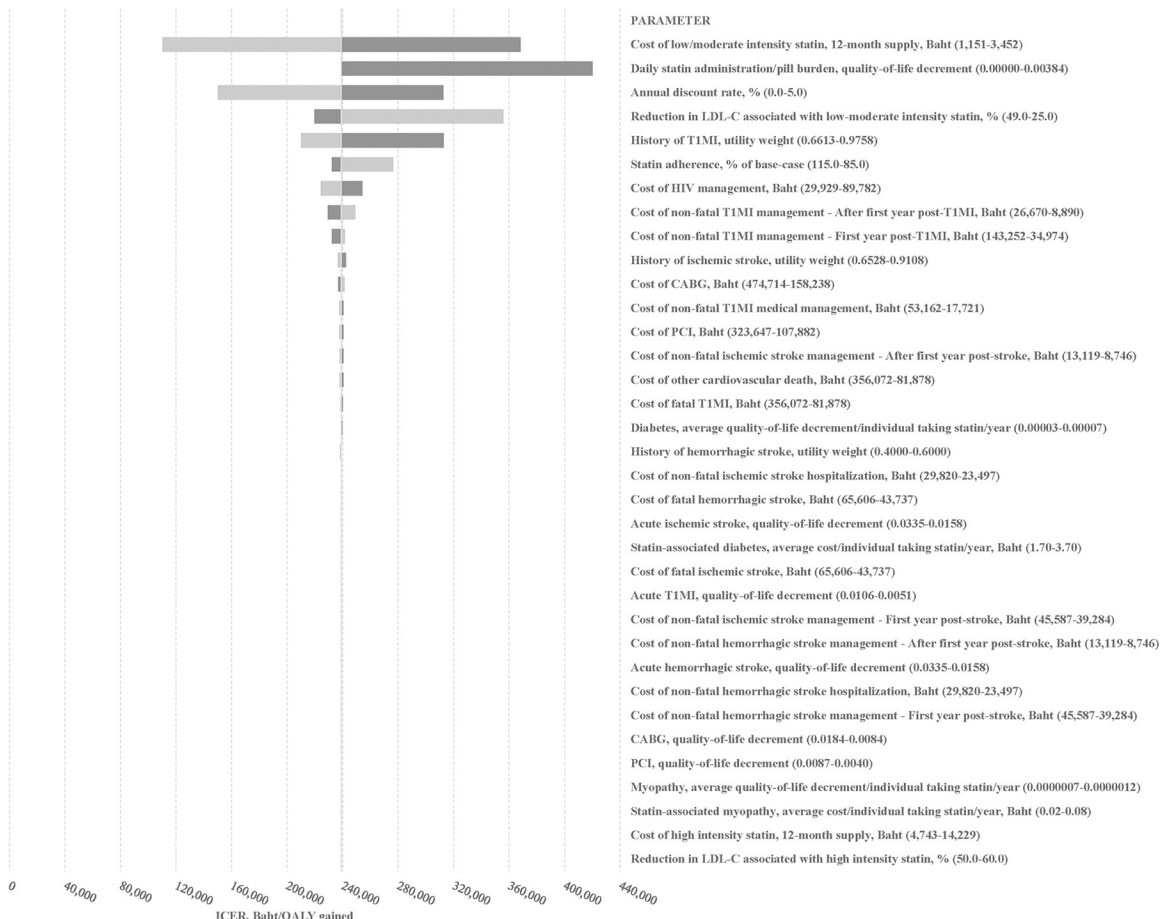

**Fig 2. Tornado plot showing impact of changes in model parameters on the incremental cost-effectiveness ratio for statin initiation threshold ≥7.5% ASCVD risk versus ≥10% ASCVD risk.** Shading of bars indicates directionality: Lighter bars represent the smaller values in the sensitivity range and darker bars indicate the larger values. Directionality also indicated by the order of values shown in the text description. Cost-effectiveness based on a willingness-to-pay threshold of 160,000 Baht/QALY gained. Baht can be converted to $US by dividing by 31.16. ASCVD, cardiovascular disease; LDL-C, low density lipoprotein cholesterol; T1MI, type 1 myocardial infarction; CABG, coronary artery bypass graft; PCI, percutaneous coronary intervention; ICER, incremental cost-effectiveness ratio; QALY, quality-adjusted life-year.

## Sensitivity analyses

One-way sensitivity analysis results are presented in Fig 2 (≥7.5% versus ≥10% ASCVD risk threshold) and S2 Fig in S1 File (≥5% versus ≥7.5% ASCVD risk threshold). Our findings were sensitive to changes in the annual cost of low/moderate intensity statin therapy, the pill burden associated with daily statin use, and the discount rate.

At a willingness-to-pay threshold of 160,000 Baht ($US5,315)/QALY gained, the ≥7.5% ASCVD risk threshold became cost-effective compared to the ≥10% ASCVD risk threshold when the average annual cost of low/moderate intensity statin dropped to 1,593 Baht ($US51; 69.2% of base-case price). This price drop could be achieved if we assumed 54% of moderate intensity atorvastatin users were instead using an equivalent dose of simvastatin (i.e., 10mg atorvastatin to 20mg simvastatin, and 20mg atorvastatin to 40mg simvastatin). Given a shift in prescribing preference to simvastatin may be possible with a shift in prescribing from protease inhibitors to integrase inhibitors, we also evaluated the price drop required for low/moderate intensity statins in the context of higher costs for HIV treatment. Assuming the maximum

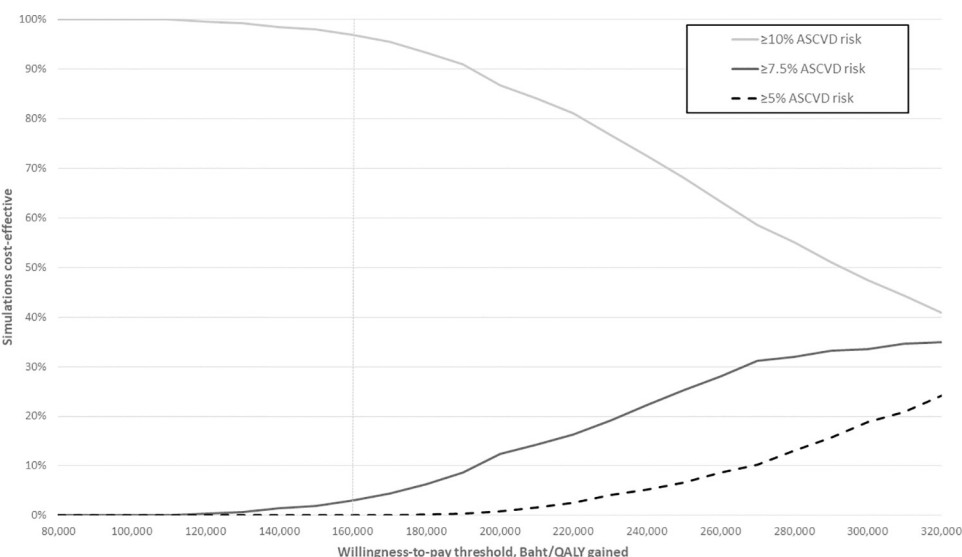

**Fig 3. Acceptability curves showing proportion of probability sensitivity analysis simulations cost-effective for each intervention under different assumptions of willingness-to-pay.** Dashed vertical line represents willingness-to-pay threshold of 160,000 Baht/QALY gained. Baht can be converted to $US by dividing by 31.16. ASCVD, atherosclerotic cardiovascular disease; QALY, quality-adjusted life-year.

annual cost of HIV management from our sensitivity range (89,784 Baht; $US2,881), the ≥7.5% ASCVD risk threshold became cost-effective compared to the ≥10% ASCVD risk threshold when the average annual cost of low/moderate intensity statin dropped to 1,459 Baht ($US47; 63.4% of base-case price). This price drop could be achieved if we assumed 64% of moderate intensity atorvastatin users were instead using an equivalent dose of simvastatin.

When the probability of ASCVD while using a statin was reduced by 30% to account for the possibility of preventative efficacy in PLHIV beyond that associated with cholesterol improvement, the ≥7.5% ASCVD risk threshold became cost-effective compared to the ≥10% ASCVD risk threshold if the average annual cost of low/moderate intensity statin use dropped to 2,085 Baht ($US66; 90.6% of base-case price).

In our base-case, we assumed that pill burden associated with daily statin use was not associated with any quality-of-life decrement. When a negative quality-of-life impact was assumed, the average number of QALYs accumulated declined with all three strategies but more so among the lower threshold strategies. At the high end of our sensitivity range, the expanded statin use strategies generated incremental QALY gains of 0.008 (ASCVD risk threshold ≥7.5% versus ≥10%) and 0.007 (ASCVD risk threshold ≥5% versus ≥7.5%) resulting in ICERs of 420,000 Baht ($US13,478)/QALY gained and 808,000 Baht ($US25,930)/QALY gained, respectively.

In probabilistic sensitivity analyses, the ASCVD risk threshold ≥10% was optimal in 96.9% of simulations at a willingness-to-pay of 160,000 Baht ($US5,135)/QALY gained, the ASCVD risk threshold ≥7.5% was optimal in 3.0% of simulations, and the ASCVD risk threshold ≥5% was optimal in 0.1% of simulations (Fig 3).

## Discussion

Reducing the ASCVD risk threshold for statin initiation would help reduce the excess risk of ASCVD among PLHIV in Thailand. However, we estimated that it would not currently be

cost-effective. Our results were sensitive to changes in statin cost but remained stable across a broad range of other sensitivity and scenario analyses.

Earlier results from Thailand and other resource-limited settings have shown statins to be an economically attractive option for primary prevention of ASCVD in the general population [19,40] However, these studies compared treatment strategies with a greater difference in efficacy than our study. For example, Tamteerano et al [19] compared statin use at various risk thresholds to a control group with no statin use. There are several other reasons our results may differ from findings in the general population: 1) there are more events competing with ASCVD in PLHIV compared with the general population [41,42]. Hence, preventing ASCVD among PLHIV results in fewer QALYs gained in comparison with preventing ASCVD in the general population; 2) background costs in PLHIV are higher than in the general population; and 3) general population studies can assume a lower cost of statin use because of the low potential for drug interactions among the general population.

Given simvastatin should not be co-administered with protease inhibitors but is safe to use with most other ART classes, reduced use of protease inhibitors is likely to stimulate greater use of simvastatin. Protease inhibitors remain in common use in Thailand (13.9% of our cohort was using a protease inhibitor), although there is strong interest towards phasing in better tolerated ART in low and middle income countries [43]. Our sensitivity analyses indicated that, in the context of increased HIV management costs, the ASCVD risk threshold ≥7.5% for statin initiation would become cost-effective if 64% of moderate intensity atorvastatin users were assumed to be using an equivalent dose of simvastatin instead. In a recent analysis of statin users at the HIV-NAT Research Collaboration/Thai Red Cross AIDS Research Centre in Bangkok, 57% of moderate intensity atorvastatin users were on protease inhibitor-based ART [14]. Therefore, replacing protease inhibitor-based ART and moderate intensity atorvastatin use with non-protease inhibitor-based ART and moderate intensity simvastatin could be sufficient to make the ASCVD risk threshold ≥7.5% cost-effective.

QALY gains in the intervention groups declined substantially when we included a small decrement in quality-of-life associated with taking a daily statin. Similarly, when a small pill burden was included in our earlier analysis of pravastatin and pitavastatin expansion, the QALY gains compared with no statin rapidly declined [8]. Current estimates of quality-of-life decrement associated with pill burden vary substantially [44,45]. However, for PLHIV, the burden of taking an additional pill along with daily ART is likely to be close to negligible.

Statin therapy typically leads to a 15–20% reduction in major ASCVD events [3]. Although, the anti-inflammatory properties of statins may increase their efficacy in PLHIV. The Randomized Trial to Prevent Vascular Events in HIV (REPRIEVE) study is currently investigating pitavastatin for the primary prevention of ASCVD in PLHIV [46]. While the results of this trial are highly anticipated, we have shown that, even if there was an additional 30% decrease in the probability of ASCVD with statin use in PLHIV, the current cost of low/moderate intensity statin use would still need to drop by 9.4% before it became cost-effective to reduce the ASCVD risk initiation threshold.

There are some limitations to this study. An analysis of the HIV Outpatient Study [47] indicated that the D:A:D equation underestimates ASCVD risk in PLHIV. However, the HIV Outpatient Study underestimates the prevalence of ASCVD family history which is an important contributor to ASCVD risk in the D:A:D equation. We also found that our main conclusions were not altered when we used the Rama-EGAT equation to calculate T1MI and stroke risk. We approximated various model parameters using data from the general population or from high income settings. It is possible these parameters differ between the general population and PLHIV, or between high income settings and the Thai healthcare setting. However, our sensitivity analyses suggested that this would not impact our main findings. Finally, we were not

able to adjust for the possible difference in ASCVD risk associated with new antiretrovirals relative to older antiretrovirals as it is not currently known.

## Conclusions

Reducing the recommended 10-year ASCVD risk threshold for statin initiation among PLHIV in Thailand would not currently be cost-effective, even if statins exhibit greater efficacy in PLHIV due to their anti-inflammatory properties. A lower threshold could become cost-effective with greater preference for simvastatin.

## Supporting information

**S1 File.**
(DOCX)

## Acknowledgments

The authors would like to acknowledge the TREAT Asia HIV Observational Database participants and Steering Committee.

## TAHOD study members

PS Ly*, V Khol, National Center for HIV/AIDS, Dermatology & STDs, Phnom Penh, Cambodia; FJ Zhang*, HX Zhao, N Han, Beijing Ditan Hospital, Capital Medical University, Beijing, China; MP Lee*, PCK Li, TS Kwong, HY Wong, Queen Elizabeth Hospital, Hong Kong SAR; N Kumarasamy*, C Ezhilarasi, Chennai Antiviral Research and Treatment Clinical Research Site (CART CRS), VHS-Infectious Diseases Medical Centre, VHS, Chennai, India; S Pujari*, K Joshi, S Gaikwad, A Chitalikar, Institute of Infectious Diseases, Pune, India; S Sangle*, V Mave, I Marbaniang, S Nimkar, BJ Government Medical College and Sassoon General Hospital, Pune, India; TP Merati*, DN Wirawan, F Yuliana, Faculty of Medicine Udayana University & Sanglah Hospital, Bali, Indonesia; E Yunihastuti*, A Widhani, S Maria, TH Karjadi, Faculty of Medicine Universitas Indonesia—Dr. Cipto Mangunkusumo General Hospital, Jakarta, Indonesia; J Tanuma*, S Oka, T Nishijima, National Center for Global Health and Medicine, Tokyo, Japan; JY Choi*, Na S, JM Kim, Division of Infectious Diseases, Department of Internal Medicine, Yonsei University College of Medicine, Seoul, South Korea; YM Gani*, NB Rudi, Hospital Sungai Buloh, Sungai Buloh, Malaysia; I Azwa*, A Kamarulzaman, SF Syed Omar, S Ponnampalavanar, University Malaya Medical Centre, Kuala Lumpur, Malaysia; R Ditangco*, MK Pasayan, ML Mationg, Research Institute for Tropical Medicine, Muntinlupa City, Philippines;

YJ Chan*, WW Ku, PC Wu, E Ke, Taipei Veterans General Hospital, Taipei, Taiwan;

OT Ng*, PL Lim, LS Lee, D Liang, Tan Tock Seng Hospital, National Centre for Infectious Diseases, Singapore (note: OT Ng is also supported by the Singapore Ministry of Health's (MOH) National Medical Research Council (NMRC) Clinician Scientist Award (MOH-000276). Any opinions, findings and conclusions or recommendations expressed in this material are those of the author(s) and do not reflect the views of MOH/NMRC.); A Avihingsanon*, S Gatechompol, P Phanuphak, C Phadungphon, HIV-NAT/Thai Red Cross AIDS Research Centre, Bangkok, Thailand; S Kiertiburanakul*, A Phuphuakrat, L Chumla, N Sanmeema, Faculty of Medicine Ramathibodi Hospital, Mahidol University, Bangkok, Thailand; R Chaiwarith*, T Sirisanthana, J Praparattanapan, K Nuket, Chiang Mai University—Research Institute for Health Sciences, Chiang Mai, Thailand; S Khusuwan*, P Kantipong, P Kambua, Chiangrai Prachanukroh Hospital, Chiang Rai, Thailand; KV Nguyen*, HV Bui, DTH

Nguyen, DT Nguyen, National Hospital for Tropical Diseases, Hanoi, Vietnam; CD Do*, AV Ngo, LT Nguyen, Bach Mai Hospital, Hanoi, Vietnam;

AH Sohn*, JL Ross*, B Petersen, TREAT Asia, amfAR—The Foundation for AIDS Research, Bangkok, Thailand; MG Law*, A Jiamsakul*, R Bijker, D Rupasinghe, The Kirby Institute, UNSW Sydney, NSW, Australia. (* TAHOD Steering Committee member)

## Author Contributions

**Conceptualization:** David C. Boettiger, Pairoj Chattranukulchai, Sasisopin Kiertiburanakul.

**Data curation:** Anchalee Avihingsanon, Romanee Chaiwarith, Suwimon Khusuwan, Matthew G. Law, Jeremy Ross, Sasisopin Kiertiburanakul.

**Formal analysis:** David C. Boettiger.

**Methodology:** David C. Boettiger, Pairoj Chattranukulchai, Anchalee Avihingsanon.

**Project administration:** David C. Boettiger.

**Supervision:** Pairoj Chattranukulchai, Sasisopin Kiertiburanakul.

**Writing – original draft:** David C. Boettiger.

**Writing – review & editing:** Pairoj Chattranukulchai, Anchalee Avihingsanon, Romanee Chaiwarith, Suwimon Khusuwan, Matthew G. Law, Jeremy Ross, Sasisopin Kiertiburanakul.

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
