## [Decision Letter · Decision Letter 0]

14 Jul 2021

PONE-D-21-09045

When to start statins for people living with HIV in Thailand: A cost-effectiveness analysis

PLOS ONE

Dear Dr. Boettiger,

Thank you for submitting your manuscript to PLOS ONE. After careful consideration, we feel that it has merit but does not fully meet PLOS ONE’s publication criteria as it currently stands. Therefore, we invite you to submit a revised version of the manuscript that addresses the points raised during the review process.

We look forward to receiving your revised manuscript.

Kind regards,

Ismaeel Yunusa, PharmD, PhD

Academic Editor

PLOS ONE

Journal Requirements:

2. Thank you for submitting the above manuscript to PLOS ONE. During our internal evaluation of the manuscript, we found significant text overlap between your submission and the following previously published work, of which you are an author.

- https://onlinelibrary.wiley.com/doi/pdf/10.1002/jia2.25494

Please revise the manuscript to rephrase the duplicated text, cite your sources, and provide details as to how the current manuscript advances on previous work. Please note that further consideration is dependent on the submission of a manuscript that addresses these concerns about the overlap in text with published work.

"DCB has received research funding from Gilead Sciences and is supported by a

National Health and Medical Research Council Early Career Fellowship

(APP1140503); MGL has received unrestricted grants from Boehringer Ingelhiem,

Gilead Sciences, Merck Sharp & Dohme, Bristol-Myers Squibb, Janssen-Cilag, and

ViiV HealthCare and consultancy fees from Gilead Sciences and data and safety

monitoring board sitting fees from Sirtex Pty Ltd; All other authors report no potential

competing interests."

Additional Editor Comments (if provided):

Please revise the manuscript by carefully considering points raised by the reviewers. Ensure the revised version also follows CHEERS reporting guidelines and re-attach the CHEERS checklist while submitting the updated manuscript.

Reviewers' comments:

Reviewer's Responses to Questions

**Comments to the Author**

1. Is the manuscript technically sound, and do the data support the conclusions?

Reviewer #1: Partly

Reviewer #2: Yes

2. Has the statistical analysis been performed appropriately and rigorously? 

Reviewer #1: N/A

Reviewer #2: Yes

3. Have the authors made all data underlying the findings in their manuscript fully available?

Reviewer #1: No

Reviewer #2: Yes

4. Is the manuscript presented in an intelligible fashion and written in standard English?

Reviewer #1: Yes

Reviewer #2: Yes

5. Review Comments to the Author

Reviewer #1: This study is a cost-effectiveness analysis of different thresholds of 10-year ASCVD risk for initiating statin use among people living with HIV (PLHIV) in Thailand. Using a microsimulation model, the authors estimated that it is not cost-effective to lower the threshold from 10% to 7.5% or 5% in the base case without reducing the cost of statins.

The manuscript is well written, and the analysis mostly follows contemporary guidelines for cost-effectiveness analyses. The findings are potentially interesting and could help guide statin prescription in the local setting. However, there are critical methodological issues that I included in the major issues section; they should be addressed to improve the study’s validity. I also included other comments and questions in the following section.

Major issues:

1. Quality-adjusted life-years (QALYs) are the claimed primary health outcome in this analysis; however, the sources for quality adjustment are from the Global Burden of Disease studies, where health outcomes are disability-adjusted life years (DALYs). QALYs and DALYs are based on different theoretical grounds and use different methodologies, hence not interchangeable (Sassi, Health Policy and Planning 2006 provides a good overview of this comparison). This error is a fundamental misconception of the difference between QALYs and DALYs. It must be corrected through either converting the ICER metric to cost per DALYs averted or finding appropriate utility weights for estimating QALYs.

2. Based on my reading of the methods, HIV progression and care are not modeled in the simulation model, and CD4 cell count remains constant for the lifetime of model individuals. Even though the study limits its population to those who have been on ART for more than six months, CD4 cell count could still increase significantly and only asymptotes after several years (Gras et al., J Acquir Immune Deﬁc Syndr 2007). Not accounting for changes in CD4 count may underestimate life expectancy (and in fact, the model trajectory of mortality was consistently higher than the mean observed values in Figure 1).

3. Figure 2: About half of the parameters have nearly zero influence on the ICERs, which is a very confusing result because these parameters should impact, in theory, either health or cost outcomes; the sensitivity analysis range for most parameters is not negligible either. Potential modeling/coding errors should be ruled out first; some explanations for this result are needed.

Other comments:

1. This study is titled “When to start statins for people living with HIV in Thailand – A cost-effectiveness analysis,” but the analysis does not concern the timing of statin initiation at all. I suggest using a more appropriate title to indicate the key strategies actually explored, i.e., CVD risk thresholds.

2. While the methods section claims that the model was calibrated to observed data, it was unclear what approach (e.g., Goodness-of-fit measure, searching algorithm) was used to calibrate the model. The calibration approach should be clearly described.

3. This study uses n = 10,000 as the size of model cohort for the microsimulation model and n = 500 for probabilistic sensitivity analysis (PSA), which is much lower than typical sizes used in microsimulation models. I am concerned about the model stability due to stochastic uncertainty and would encourage the authors to use n = 100,000 for the model cohort and n = 1,000 for the PSA.

4. Figure 1: It would be helpful to provide the uncertainty interval from the model trajectories as well – It helps address concerns on model stability as well.

5. The methods section claims that quality-of-life adjustment for the disutility of pill-taking was not considered because the population is already required to take ART pills, which is a reasonable assumption. However, the results indicate that this disutility was indeed explored in their analysis, inconsistent with the methods description. I suggest rewording the methods to frame this disutility as a sensitivity analysis to make the flow consistent.

6. Following the rationale of not including the disutility of pill-taking due to ART use, I wonder if statin adherence should be assumed to be equal to rates observed in the general population. It may be higher because of the exact reason (no added disutility because PLHIV on ART are required to take daily pills already). It would be interesting to discuss this topic since adherence is a critical factor in statin use guidelines.

7. The utility weight for those without a history of CVD was set at 1, which is too high considering this is an HIV-positive population. The authors should first fix the misuse of DALY weights for estimating QALYs, and if they decide to switch to DALYs as the health outcomes, GBD estimates could be used for this value. For example, GBD 2016 estimated a disability weight of 0.078 for PLHIV on ART.

8. Scenario analyses: What is the rationale for using this alternative Rama-EGAT equation as a scenario analysis? The Rama-EGAT equation was developed from an HIV-negative population and was not validated in PLHIV.

Reviewer #2: This is a well written article with nice statistical analysis. It would have been worthwhile to use widely accepted ASCVD risk calculation from AHA/ACC for sensitivity analysis including only people with >40years.

6. PLOS authors have the option to publish the peer review history of their article (what does this mean?). If published, this will include your full peer review and any attached files.

Reviewer #1: No

Reviewer #2: **Yes: **Hyun Joon Shin

---

## [Author Response · Author response to Decision Letter 0]

16 Aug 2021

PLOS ONE

August 11th, 2021

Re. Atherosclerotic cardiovascular disease thresholds for statin initiation among people living with HIV in Thailand: A cost-effectiveness analysis 

(Formerly: When to start statins for people living with HIV in Thailand: A cost-effectiveness analysis)

We hereby re-submit the above manuscript to be considered for publication by PLOS ONE. Detailed responses to reviewer and editor comments are provided below. 

Editor

1. Please revise the manuscript by carefully considering points raised by the reviewers. Ensure the revised version also follows CHEERS reporting guidelines and re-attach the CHEERS checklist while submitting the updated manuscript.

Response: Revised CHEERS checklist included with resubmission. 

Reviewer #1

1. Quality-adjusted life-years (QALYs) are the claimed primary health outcome in this analysis; however, the sources for quality adjustment are from the Global Burden of Disease studies, where health outcomes are disability-adjusted life years (DALYs). QALYs and DALYs are based on different theoretical grounds and use different methodologies, hence not interchangeable (Sassi, Health Policy and Planning 2006 provides a good overview of this comparison). This error is a fundamental misconception of the difference between QALYs and DALYs. It must be corrected through either converting the ICER metric to cost per DALYs averted or finding appropriate utility weights for estimating QALYs.

Response: Thank you for this comment. DALY estimates in the GBD are based on a comprehensive list of utility weights for different health states. These weights can be used to calculate both DALYs and QALYs. We used utility weights defined in the GBD to calculate QALYs. This is common practice in health economics literature (e.g., Kazi 2014 JAMA 316(7):743, Heller 2017 Circulation 136:1087, Boettiger 2021 JIAS 24(3):e25690).

2. Based on my reading of the methods, HIV progression and care are not modeled in the simulation model, and CD4 cell count remains constant for the lifetime of model individuals. Even though the study limits its population to those who have been on ART for more than six months, CD4 cell count could still increase significantly and only asymptotes after several years (Gras et al., J Acquir Immune Deﬁc Syndr 2007). Not accounting for changes in CD4 count may underestimate life expectancy (and in fact, the model trajectory of mortality was consistently higher than the mean observed values in Figure 1).

Response: Thank you. This is an important observation about an aspect of our model that we spent many months discussing. It is now recognized that people with well controlled HIV have a normal life expectancy (Edwards 2021 Ann Int Med). As the reviewer points out, CD4 count may take several years to asymptote, however, after 6mths of ART the vast majority of PLHIV have a CD4 count >500 cells/mm3 – indicative of a strong immune system. We therefore believe it was reasonable to assume no clinically meaningful change in CD4 count beyond 6mths of ART initiation. The reason our modelled mortality rate was slightly higher than the observed mortality rate is most likely due to underreporting of death in TAHOD associated with patients being reported as lost to follow up rather than having died.

3. Figure 2: About half of the parameters have nearly zero influence on the ICERs, which is a very confusing result because these parameters should impact, in theory, either health or cost outcomes; the sensitivity analysis range for most parameters is not negligible either. Potential modeling/coding errors should be ruled out first; some explanations for this result are needed.

Response: It is interesting that many of our parameters had little influence on the final ICERs. This was not unexpected. These parameters are mostly costs or utility weights that are incurred for very short-term and/or rare events, such as acute stroke, MI, and statin-associated diabetes. To confirm the low likelihood of any coding/modelling errors, we have checked our code/model again. We did not isolate anything requiring correction.

4. This study is titled “When to start statins for people living with HIV in Thailand – A cost-effectiveness analysis,” but the analysis does not concern the timing of statin initiation at all. I suggest using a more appropriate title to indicate the key strategies actually explored, i.e., CVD risk thresholds.

Response: Thank you. We have modified the title to: “Atherosclerotic cardiovascular disease thresholds for statin initiation among people living with HIV in Thailand: A cost-effectiveness analysis”

5. While the methods section claims that the model was calibrated to observed data, it was unclear what approach (e.g., Goodness-of-fit measure, searching algorithm) was used to calibrate the model. The calibration approach should be clearly described.

Response: We thank the reviewer for this suggestion and have clarified that our calibration procedure used a goodness-of-fit approach. On pg5 it states: “We calibrated our model using a goodness-of-fit approach based on the observed rates of all-cause and cardiovascular death among TAHOD participants between 2009 and 2019. Figure 1 shows that our calibrated model estimates provided an accurate reflection of the observed data.”

6. This study uses n = 10,000 as the size of model cohort for the microsimulation model and n = 500 for probabilistic sensitivity analysis (PSA), which is much lower than typical sizes used in microsimulation models. I am concerned about the model stability due to stochastic uncertainty and would encourage the authors to use n = 100,000 for the model cohort and n = 1,000 for the PSA.

Response: We chose n=10,000 as our model estimates were stable with this cohort size. However, we agree with the reviewer that n=100,000 and n=1,000 are more in line with current practice. We have therefore re-run our models using these numbers. The results were closely aligned with our original findings.

7. Figure 1: It would be helpful to provide the uncertainty interval from the model trajectories as well – It helps address concerns on model stability as well.

Response: We respectfully disagree with this suggestion. The model uncertainty is a function of the number of microsimulations we choose to run. Providing a confidence interval around our model trajectories would misleadingly imply we do not have control over what the confidence interval looks like.

8. The methods section claims that quality-of-life adjustment for the disutility of pill-taking was not considered because the population is already required to take ART pills, which is a reasonable assumption. However, the results indicate that this disutility was indeed explored in their analysis, inconsistent with the methods description. I suggest rewording the methods to frame this disutility as a sensitivity analysis to make the flow consistent.

Response: Thank you for picking up on this. We have revised the wording in the methods to reflect that no pill burden was a key assumption in our base-case model: “Since patients using ART are already required to take at least one daily pill, our base-case model assumed that remembering to take a daily statin and the inconvenience of doing so (pill burden) was not associated with a quality-of-life decrement.” (p7)

9. Following the rationale of not including the disutility of pill-taking due to ART use, I wonder if statin adherence should be assumed to be equal to rates observed in the general population. It may be higher because of the exact reason (no added disutility because PLHIV on ART are required to take daily pills already). It would be interesting to discuss this topic since adherence is a critical factor in statin use guidelines.

Response: Our estimates of statin adherence were based on a 2020 study we conducted in Thailand among PLHIV (Boettiger, 2020, AIDS, Maintenance of statin therapy among people living with HIV). As the reviewer suggests, this study found that statin adherence appears to be slightly better among PLHIV than in the general population.

10. The utility weight for those without a history of CVD was set at 1, which is too high considering this is an HIV-positive population. The authors should first fix the misuse of DALY weights for estimating QALYs, and if they decide to switch to DALYs as the health outcomes, GBD estimates could be used for this value. For example, GBD 2016 estimated a disability weight of 0.078 for PLHIV on ART.

Response: We agree with the reviewer that PLHIV on ART do not have the same QOL as the general population. However, the general population are not included in our model. The best-case health scenario for those included was to be HIV-positive and free of CVD. Therefore, we believe it is appropriate to apportion this group a utility weight of 1, indicating optimal health for the population being studied. 

11. Scenario analyses: What is the rationale for using this alternative Rama-EGAT equation as a scenario analysis? The Rama-EGAT equation was developed from an HIV-negative population and was not validated in PLHIV.

Response: We agree that Rama-EGAT, as a risk equation based on the general population, is not the ideal ASCVD equation to be used for our model. However, it is widely used for PLHIV in Thailand and well regarded by Thai physicians. It is also not uncommon for clinical guidelines to recommend general population ASCVD equations for PLHIV (e.g., Grundy 2018 JACC 73(24):3168-3209, and Royal College of Physicians of Thailand 2016 Clinical Practice Guideline on Pharmacologic Therapy of Dyslipidemia for ASCVD prevention http://www.thaiheart.org/Download/2016-RCPT-Dyslipidemia-Guideline.html). We therefore believe it was reasonable to evaluate Rama-EGAT as a scenario analysis. 

Reviewer #2

1. This is a well written article with nice statistical analysis. It would have been worthwhile to use widely accepted ASCVD risk calculation from AHA/ACC for sensitivity analysis including only people with >40years. 

Response: Thank you for this comment. The DAD equation is a widely accepted, HIV-specific ASCVD risk equation modelled off the Framingham equation and recommended by the AHA/ACC (Feinstein 2019 Circulation 139:e1-e27). In response to this reviewer comment, we have run our model for a population >40 years old (as opposed to >35y in the original analysis). The results are shown in our cover letter. Given these are of a similar magnitude to our base-case, we have not added these findings to the revised manuscript. 

Competing interests and data availability

DCB has received research funding from Gilead Sciences and is supported by a National Health and Medical Research Council Early Career Fellowship (APP1140503); MGL has received unrestricted grants from Boehringer Ingelhiem, Gilead Sciences, Merck Sharp & Dohme, Bristol-Myers Squibb, Janssen-Cilag, and ViiV HealthCare and consultancy fees from Gilead Sciences and data and safety monitoring board sitting fees from Sirtex Pty Ltd; All other authors report no potential competing interests. These declarations do not alter our adherence to PLOS ONE policies on sharing data and materials. Data were collected as part of a regional cohort collaboration. The cohort collaboration has data-sharing policies that were approved by the corresponding IRB and specify that both internal and external investigators are subject to a formal process to request access to the data through submission of a concept sheet that adheres to these policies. This study was conducted under these policies, and data will only be available upon request for researchers who meet the criteria for access to confidential data. Interested individuals should contact Boondarika Petersen (tor.nakornsri@treatasia.org).

Please contact me at dboettiger@kirby.unsw.edu.au should you have any questions.

Sincerely

Dr David Boettiger, PhD

---

## [Editor Report · Decision Letter 1]

19 Aug 2021

Atherosclerotic cardiovascular disease thresholds for statin initiation among people living with HIV in Thailand: A cost-effectiveness analysis

PONE-D-21-09045R1

Dear Dr. Boettiger,

We’re pleased to inform you that your manuscript has been judged scientifically suitable for publication and will be formally accepted for publication once it meets all outstanding technical requirements.

Kind regards,

Ismaeel Yunusa, PharmD, PhD

Academic Editor

PLOS ONE
---

## [Editor Report · Acceptance letter]

25 Aug 2021

PONE-D-21-09045R1 

Atherosclerotic cardiovascular disease thresholds for statin initiation among people living with HIV in Thailand: A cost-effectiveness analysis 

Dear Dr. Boettiger:

I'm pleased to inform you that your manuscript has been deemed suitable for publication in PLOS ONE. Congratulations! Your manuscript is now with our production department. 

Kind regards, 

on behalf of

Dr. Ismaeel Yunusa 

Academic Editor

PLOS ONE